# Comparison between Lipase Performance Distributed at the O/W Interface by Membrane Emulsification and by Mechanical Stirring

**DOI:** 10.3390/membranes11020137

**Published:** 2021-02-16

**Authors:** Emma Piacentini, Rosalinda Mazzei, Lidietta Giorno

**Affiliations:** National Research Council, Institute on Membrane Technology, CNR-ITM, Via P. Bucci 17 C, 87036 Rende, Italy; r.mazzei@itm.cnr.it

**Keywords:** multiphasic bioreactor systems, membrane emulsification, lipase, kinetic resolution, emulsion, enantiomer

## Abstract

Multiphase bioreactors using interfacial biocatalysts are unique tools in life sciences such as pharmaceutical and biotechnology. In such systems, the formation of microdroplets promotes the mass transfer of reagents between two different phases, and the reaction occurs at the liquid–liquid interface. Membrane emulsification is a technique with unique properties in terms of precise manufacturing of emulsion droplets in mild operative conditions suitable to preserve the stability of bioactive labile components. In the present work, membrane emulsification technology was used for the production of a microstructured emulsion bioreactor using lipase as a catalyst and as a surfactant at the same time. An emulsion bioreaction system was also prepared by the stirring method. The kinetic resolution of (*S*,*R*)-naproxen methyl ester catalyzed by the lipase from *Candida rugosa* to obtain (*S*)-naproxen acid was used as a model reaction. The catalytic performance of the enzyme in the emulsion systems formulated with the two methods was evaluated in a stirred tank reactor and compared. Lipase showed maximum enantioselectivity (100%) and conversion in the hydrolysis of (*S*)-naproxen methyl ester when the membrane emulsification technique was used for biocatalytic microdroplets production. Moreover, the controlled formulation of uniform and stable droplets permitted the evaluation of lipase amount distributed at the interface and therefore the evaluation of enzyme specific activity as well as the estimation of the hydrodynamic radius of the enzyme at the oil/water (o/w) interface in its maximum enantioselectivity.

## 1. Introduction

Many interesting biocatalytic reactions involve organic compounds that are poorly soluble in water. These reactions are carried out in biphasic systems. A biphasic bioreactor may consist of a dispersed organic phase in the form of droplets (containing the hydrophobic substrate to be converted), a continuous aqueous phase (in which the reaction product is extracted), and a phase transfer biocatalyst displaced at the organic/water interface. Lipases are the most used enzymes in synthetic organic chemistry, catalyzing the chemo-, regio- and/or stereo-selective hydrolysis of esters or the reverse reaction in organic solvents [1,2,3,4,5,6]. Biotechnological applications of lipases in the synthesis of many organic molecules in non-aqueous media have rapidly increased [7,8,9,10,11,12,13,14]. One of the most important characteristics of lipases is their activation by oil–water interfaces (interfacial activation) [15,16]. Lipases have only a marginal activity toward substrates in a monomeric state but show a high activity when the substrate concentration is high enough to form micellar aggregates or emulsions.

Many applications in two-phase media use a stirred tank bioreactor where high-energy inputs are required to obtain a complete and uniform dispersion of the two phases. This usually causes enzyme deactivation during the process. In fact, it has been observed that various enzymes lose activity when the solution is maintained under a high shear stress for a long time [17,18,19,20]. Moreover, droplet size distribution and the interfacial area are continually changing, and phase separation is observed when the agitation is stopped. Sometimes, it is necessary to add emulsifiers to stabilize the emulsion [21].

Over the last 30 years, there has been a growing interest in membrane emulsification [22,23,24,25,26,27]. The distinguishing feature is that droplet size is controlled by the choice of the pore size of a microporous membrane. The dispersed phase is pressurized through the membrane pores and detached drop-by-drop at the interface between the membrane pore opening and the continuous phase. The benefits of membrane emulsification for phase transfer biocatalysis could derive from the low shear stress applied and the formation of fine droplets with uniform size distribution. The biocatalyst can be directly distributed at the organic–water interface due to their physicochemical characteristics exhibiting simultaneously emulsifying and catalytic properties. In our previous work, the suitability of membrane emulsification to distribute the lipase at the oil/water (o/w) interface was demonstrated [28]. Here, we further prove the superior stability of emulsion droplets and higher enantioselectivity of lipase distributed at the interface of oil/water emulsions prepared by membrane emulsification compared to emulsions prepared by mechanical stirring. The aim of the work is to demonstrate the power of membrane emulsification technology to implement phase transfer catalysis applications.

The enantioselective hydrolysis of racemic naproxen methyl ester catalyzed by lipase from *Candida rugosa* was carried out. The dispersed phase contained the hydrophobic substrate, i.e., racemic naproxen methyl ester, whilst the hydrophilic product, i.e., the naproxen acid, was extracted in the aqueous phase. The hydrophobic substrate can be converted with high yield, because the removal of the water-soluble product from the reaction microenvironment to the aqueous phase can shift the thermodynamic equilibrium of the reactions on the basis of the Le Chatelier principle [29,30]. In this system, the interface provided stable reaction conditions, because the interfacial emulsion area produced by the membrane emulsification process did not change during the course of the reaction. The stability of the interface was assured by uniform and small droplets obtained by the membrane emulsification process. The uniform and small droplets permitted obtaining a stable and large interface for the conversion of hydrophobic substrate.

In addition to the membrane emulsification process, a conventional mechanical method was used to produce oil-in-water emulsion for the enzymatic hydrolysis of naproxen ester. In both systems, the reaction was carried out by using the same enzyme and substrate concentration, at the same temperature and oil/water ratio. To critically evaluate the performance of the two methods used to distribute the enzyme at the oil/water interface, the following parameters were considered: enantioselectivity, conversion of the substrate, and enzyme specific activity. In addition, the production of emulsions by membrane emulsification permitted the evaluation of molecular properties of lipase distributed at the o/w interface. In fact, thanks to the droplets stability, it was possible to evaluate the molecular diameter of lipase at the emulsion interface on the basis of easily measurable parameters, such as droplet size, dispersed phase volume, and mass of the enzyme at the interface.

## 2. Materials and Methods

### 2.1. Chemicals 

Lipase from *Candida Rugosa* (Sigma Aldrich, Milan, Italy), type VII, 1.180 units/mg solid was used. The protein was dissolved in 50 mM phosphate buffer solution (pH 7) prepared with ultrapure water, sodium dihydrogen phosphate anhydrous (NaH_2_PO_4_), and disodium hydrogen phosphate anhydrous (Na_2_HPO_4_) provided by Sigma Aldrich (Milan, Italy). Before its use, the lipase solution was centrifuged at 3000 rpm for 15 min. Racemic naproxen methyl ester was used as substrate after it was dissolved in isooctane (Sigma Aldrich, Milan, Italy). Pure (S)-naproxen isomer obtained from Sigma Aldrich (Milan, Italy) was used as a standard for chiral HPLC analyses. THF (tetrahydrofuran) and TEA (triethylamine) HPLC grade were obtained from Sigma Aldrich. Methanol HPLC grade were obtained from Carlo Erba, Milan, Italy.

### 2.2. Membrane and Membrane Emulsification System

Emulsification experiments in crossflow mode were carried out in a homemade lab-scale emulsification plant, as schematized in Figure 1. The membrane used was a tubular shirasu porous glass membrane (SPG) membrane (supplied from SPG Technology, Miyazaki, Japan) having a mean pore size of 0.1 μm, outer diameter of 10 mm, and length of 100 mm. The membrane tube was installed inside a laboratory made glass module.

The oil phase was circulated along the shell side of the SPG membrane and forced to permeate through the membrane by applying a transmembrane pressure of about 0.1 bar. The continuous phase flows along the lumen membrane surface at a fixed axial velocity. When the desired dispersed phase volume was permeated in the continuous phase, the dispersed phase circuit was depressurized to stop the oil permeation, and the obtained emulsion was collected and kept in a glass beaker for further characterization and for the reaction hydrolysis of (*R*, *S*) naproxen methyl ester.

### 2.3. The Geometry of the Stirred Tank Reactor (STR)

A stirred tank reactor designed and developed in our laboratory to achieve complete mixing during the reaction was used. Its dimensions consider both mixing effectiveness and structural considerations. The mechanically stirred tank bioreactor used in the present work is shown in Figure 2, and its relative dimensions are reported in Table 1.

### 2.4. Experimental Procedure

#### 2.4.1. Emulsion Bioreaction Carried Out Using the Lipase Distributed at the Oil/Water Interface by Membrane Emulsification (ME)

The o/w emulsion was prepared using a solution of methyl ester of naproxen in isooctane as the dispersed phase and 100 mL of lipase 2 g/L in 50 mM phosphate buffer (pH 7) as the continuous phase. The total volume of the dispersed phase permeated in the continuous phase was 11.56 mL, and 10.36% of organic phase in total volume emulsion was obtained. The transmembrane pressure applied to permeate the dispersed phase through the membrane pore was 0.1 bar, and the average flux was 5.4 L/hm^2^. A continuous phase axial velocity of 0.3 m/s was used to detach the emulsion droplets formed at the membrane pore level. The emulsification time was 48 min, and the process was stopped when the volume of organic phase in the emulsion was equivalent to the maximum volume of the organic phase that the lipase was able to disperse in the aqueous phase (11.56 mL) without observe phases separation. The emulsion was collected in a glass beaker where the reaction was continued under gentle stirrer (just to maintain a uniform dispersion of the emulsion) and thermostated at 30 °C (temperature optimum [12,13]) for 24 h. During the reaction time, at appropriate intervals, samples were taken from the reaction vessel and the concentrations of (S)- and (R)-isomer were measured by chiral HPLC. Samples from the aqueous phase of the emulsion were obtained by filtering a small amount of emulsion through 0.2 µm cellulose acetate membranes. In addition, the samples were analyzed by SDS-PAGE electrophoresis in order to evaluate the presence of lipase not distributed at the emulsion interface and that which eventually remained in the water continuous phase.

#### 2.4.2. Emulsion Bioreaction Carried Out Using the Lipase Distributed at the Oil/Water Interface by Stirring Tank Reactor (STR)

First, 23 mL of 50 mM phosphate buffer (pH 7.0) containing 2 g/L of lipase (raw weight) and 2.66 mL of racemic naproxen methyl ester solution in isooctane (10.36% organic phase in emulsion) were stirred at 950 rpm (0.9 m/s) in the stirred tank reactor (STR). The stirring rate value was selected in order to achieve a complete distribution as a form of droplets of the organic phase in the aqueous phase. A thermostatic bath was used to keep the temperature constant at 30 °C. The reaction was carried out for 24 h, and samples of 0.5 mL were collected at appropriate intervals. To separate the naproxen methyl ester present in the organic phase, from the naproxen acid present in the aqueous phase, the sample was filtered through a 0.2 μm cellulose acetate filter. In this way, the reaction was also stopped by removing the substrate. The (*R*)- and (*S*)-naproxen in the aqueous phase were measured by chiral HLPC. 

### 2.5. Measurement of (R)- and (S)-Naproxen Acid in the Aqueous Phase

Concentration of (*S*)-naproxen acid and its isomer, (*R*)-naproxen, was measured by HPLC (Merk Hitachi, Tokyo, Japan) coupled with an UV detector (Merk Hitachi L-4000) at 254 nm. The column was a CHIROBIOTIC V (Astec, Chattanooga, TN, USA). The mobile phase was a mixture 10/90 of THF/aqueous solution at pH 7.0 containing 0.1% TEA. The analysis was made at a flow rate of 1.0 mL/min. (*S*)-naproxen acid was eluted faster than (*R*)-naproxen acid. The calibration curve of chromatographic peak area versus naproxen concentration was obtained from standards solutions.

The overall error associated with analyses was about 0.2%, which was calculated over 15 subsequent injections and monitored by injecting one standard every 10 analyzed samples.

### 2.6. Emulsion Characterization

Emulsion droplets were observed by optical microscopy (Zeiss, Jena, Germany, model Axiovert 25), equipped with a camera (JVC, Yokohama, Japan, model TK-C1481BEG) to capture the images of the emulsion. Pictures were analyzed by the Scion Image program that allows the automatic counting and measurement of the droplets present in a selected area. From these measurements, the mean droplet size and size distribution were evaluated. Three samples were analyzed for each experiment, and the reported results are the average of three different experiments.

The mean particle size was expressed as the surface weighted mean diameter (or Sauter diameter), *D*[3,2] and as the volume weighted mean diameter, (or De Brouckere diameter), *D*[4,3]. *D*[3,2] and *D*[4,3] were determined, respectively, as follows:(1)$$D\left[3,2\right]=\frac{\sum D_{i}^{3}n_{i}}{\sum D_{i}^{2}n_{i}}$$
(2)$$D\left[4,3\right]=\frac{\sum D_{i}^{4}n_{i}}{\sum D_{i}^{3}n_{i}}$$
where *D_i_* = particle diameter of class *i* and *n_i_* = number of particles in class *i*.

The width of droplet size distribution was expressed as a Span number, which is calculated by the following expression:(3)$$Span=\frac{D\left[90\right]-D\left[10\right]}{D\left[50\right]}$$
where *D*[*x*0] is the diameter corresponding to *x*0 vol % on a relative cumulative droplet size curve.

### 2.7. Electrophoresis

Proteins present in the collected samples, e.g., lipase in the initial solution and in the aqueous phase after filtration of emulsion samples (un-adsorbed lipase proteins), were analyzed by one-dimensional SDS-PAGE on a 10–15% PhastGel^TM^ gradient using buffer strips. An 8/1 μL sample applicator was used (Amersham Biosciences, Little Chalfont, UK). The gel has a continuous 10–15% PastGel^TM^ gradient gel with 2% crosslinking using buffer strips composed of 0.20 M Tris-glycine, 0.20 M Tris, and 0.55% SDS at pH 8.1.

Sample treatment: To the final volume, 2.5% SDS (Sigma-Aldrich, Milan, Italy) and 5% *β*-mercaptoethanol (Sigma-Aldrich, Milan, Italy) were added and heated at 100 °C, and then, 0.01% of bromophenol blue (Sigma-Aldrich, Milan, Italy) was added. Each sample was loaded onto a separate lane of the gel containing 1 μL of sample. The gels were stained with silver (Silver Staining kit, Protein, GE Healthcare, Chicago, IL, USA) and then destained with 3.7% Tris-HCl, and 1.6% sodium thiosulfate. The solution for preserving the gels contained 10% glycerol.

The gel images captured by scanner were analyzed by Image Quant TL Software (Amersham Biosciences, Little Chalfont, UK), which permitted identifying band molecular weights (MW) and concentration. The estimation of protein MW was calculated by using the molecular size calibration mode in a gel image containing standard MWs. The MWs of the proteins of unknown samples were calculated from the logarithmic curve fitting, which relates the standard MWs with the relative mobility as pixel position by using calibration Kit proteins (LMW, low molecular weight). The amounts of proteins identified in the gels were calculated from the quantitative calibration curve, using standard proteins, which relates the band volume and intensity to protein amount.

### 2.8. Enantiomeric Excess, Substrate Conversion, Specific Activity Measure, and Analysis of Shear Stress Condition

The enantiomeric excess of (*S*)-isomer in the product, *ee_p_* %, was calculated according to:(4)$$ee_{p}=\frac{\left(S_{p}-R_{p}\right)}{\left(S_{p}+R_{p}\right)}\times 100$$
where *S_p_* and *R_p_* are the mass of (*S*)- and (*R*)-naproxen acid produced, respectively.

Conversion of substrate was calculated according to:(5)$$Conversion=\frac{\left(S_{p}+R_{p}\right)}{M_{s}}\times 100$$
where *M_s_* is the mass of the racemic substrate.

The specific activity is the amount of (*S*)- and (*R*)-naproxen acid produced in a given amount of time under given conditions per milligram of enzyme (mmol mg^−1^ h^−1^).

### 2.9. Evaluation of Diameter of the Lipase at the Interface

The precise control of emulsion production by the membrane emulsification process permitted also the measurement of the diameter of the enzyme when it was distributed at the emulsion interface. In fact, if we know the total surface area of the droplets and the total number of lipase molecules distributed at the interface, assuming that the enzyme fully covers uniformly the surface of the droplets, then it is possible to calculate the hydrated diameter of the enzyme at the interface.

Briefly, the total surface area of spherical drops can be calculated by the total number of drops (*N_tot_*) and the mean surface area of individual drop (*A_d_*):$$A_{Tot}=N_{Tot}\;A_{d}.$$

Here, $A_{d}=4\pi r_{d}^{2}$ (*r_d_* is the radius of the drop), and
$$N_{Tot}=\frac{V_{Tot}}{V_{d}}$$
where *V_Tot_* is the total volume contained in the drops, which corresponds to the organic phase permeated in the aqueous phase, and *V_d_* is the mean volume of a single spherical drop ($V_{d}=\;\frac{4\;\pi \;r_{d}^{3}}{3}$).

The total number of lipase molecules (*N_Tot_*_. lipase_) can be calculated by the lipase mass distributed at the emulsion interface ($m_{lipase}$)
$$N_{Tot.\;\mathrm{lipase}}=\;N_{A}\;\frac{m_{lipase}}{MW_{lipase}}$$
where $N_{A}$ is the Avogadro number and $MW_{lipase}$ is the lipase molecular weight.

Dividing the total surface area of the droplets by the total number of lipase macromolecules distributed at the interface, the area that each lipase molecule occupies at the interface is obtained:$$A_{lipase}=\frac{A_{Tot}}{N_{Tot.\;\mathrm{lipase}}}.$$

Approximating the lipase to a spherical macromolecule (Figure 3), the area that a single lipase macromolecule occupied at the o/w interface of a spherical oil drop corresponded to the area of the circle identified by the intersection of the spherical macromolecule with the surface of the oil drop ($A_{circle\;occupied\;by\;lipase\;at\;interface}=\pi r_{lipase\;at\;interface}^{2}$), from which the hydrodynamic macromolecular diameter of lipase at the interface can be obtained.

## 3. Results

### 3.1. Emulsion Size Distribution

In the present work, an oil-in-water emulsion was prepared by membrane emulsification and by mechanical stirring without surfactants, which are demonstrated to cause enzyme inactivation due to structural changes of the enzyme. Emulsion size distribution could not be determined when the emulsion was prepared by the mechanical stirring method, because phase separation occurred as soon as the stirring was stopped. On the contrary, a stable emulsion was produced by membrane emulsification, and its size distribution is shown in Figure 4.

*D*[3,2] and *D*[4,3] of 1.49 (±0.3) µm and 1.52 (±0.3) µm were obtained, respectively while the span was 0.2. The emulsion was stable without the addition of other surfactants, and no phase separation was observed when the emulsification process was stopped. The size of the emulsion was evaluated after 24 h from the end of the emulsification process and any change in droplet size distribution was observed.

### 3.2. Performance of Lipase at the Emulsion Interface Prepared by Membrane Emulsification and Mechanical Stirring

In Figure 5, the conversion (%) and the enantiomeric excess (*ee_p_*) (%) for the emulsion reaction system prepared by membrane emulsification and mechanical stirring are compared.

When the enantioselective hydrolysis of racemic naproxen methyl ester was carried out in a stirred tank reactor that also prepared the emulsion (STR), a conversion of around 60% was achieved in 24 h. Then, 100% of *ee_p_* was obtained as a function of the reaction time until the conversion was slightly higher than 40%. However, when the conversion further increased, (*R*)-naproxen acid was also produced, and the *ee_p_* percentage decreased down to 75%. If the emulsion to distribute the lipase at the interface was produced by the membrane emulsification process, a conversion of around 50% was achieved in 24 h, and 100% of *ee_p_* was maintained until a total conversion of (*S*)-naproxen methyl ester was reached. The stable and uniform droplets emulsion produced by membrane emulsification permitted obtaining a maximum conversion of (*S*)-naproxen ester in (*S*)-naproxen acid without reducing the enantiomeric excess, demonstrating a high enantioselective interaction between the enzyme and the hydrophobic substrate. Data indicated that the manufacturing of emulsion interface through the membrane technology created a favorable microenvironment for the interaction with the hydrophobic substrate and the extraction of the hydrophilic product and a stable lipase conformation that permitted enantioselective kinetics in mild operative conditions throughout the entire reaction time. On the contrary, the mechanical shear and the uncontrolled production of emulsion droplets in the STR determined a reduction in the enantioselective properties of the enzyme so that lipase was able to also interact with the (*R*)-naproxen methyl ester and convert it into the corresponding^®^-naproxen acid.

When the membrane emulsification method was used to produce the emulsion, first, the specific activity decreased and became constant (Figure 6) just after the emulsion production process was completed (about 48 min). After that time, a constant enzyme amount was measured (0.012 g), demonstrating that the precise control of droplets size permitted having the same enzyme amount for the remaining reaction time, which worked with almost constant specific activity. The monodispersed emulsion guarantees a constant reservoir of substrate volume at the beginning of the reaction due to emulsion stability (see Section 3.1). On the contrary, when the emulsion was produced by the stirring method, the specific activity decreased throughout the reaction time, even though with an initial less steep slope. This is probably due to the continuous destabilization of the interface (emulsion rupture/formation), which did not permit having a constant enzyme amount and a constant reservoir of substrate. The same trend was achieved for the catalytic activity (Figure 6) in the two systems. The decrease in the catalytic activity and specific activity in both systems may also depend on the decrease of substrate concentration due to the continuous conversion of the racemic mixture as a function of time. In Table 2, the mean specific activity and the corresponding conversion percentage and *ee_p_* percentage for the two systems after 24 h were compared.

Due to the higher enantioselectivity, the conversion and the specific activity of the lipase distributed at the emulsion interface by membrane emulsification were obviously lower compared to the lipase distributed at the interface by the stirring method. In fact, in the former case, being able to interact only with the (*S*)-naproxen, lipase worked with a lower substrate concentration compared to the latter case, which could interact also with the (*R*)-naproxen. However, a similar specific activity with a maximum conversion of (*S*)-naproxen ester in (*S*)-naproxen acid was achieved after 24 h without any reduction of enantiomeric excess just when membrane emulsification was used to prepare the emulsion. When the lipase was distributed at the interface using the stirring method, the mechanical shear conditions applied and the uncontrolled production of emulsion droplets caused a reduction in the enantioselective properties of enzyme (to 75.2% after 24 h). A higher axial velocity (0.9 m/s) was used in the STR compared with the membrane emulsification process (0.3 m/s) in order to maintain the dispersion of the two phases, which correspond to a higher mechanical stress for the enzyme. Results indicated that enzymes with interfacial properties such as lipases can have high catalytic activity and high enantioselectivity when working in proper conditions of microenvironment reaction created by the membrane process.

### 3.3. Evaluation of the Hydrodynamic Radius of Lipases at the o/w Interface

As shown in Figure 7, there was no lipase in the recovered aqueous solution; therefore, it can be assumed that the entire lipase amount present in the continuous phase was distributed at the interface of the stable emulsion prepared using the membrane process.

In control experiments, (i.e., by circulating a protein solution along the lumen circuit, in absence of emulsion, at the same axial velocity—0.9 m/s—used for the continuous phase), it was also verified that no lipase adsorbed to the membrane (i.e., the lipase solution concentration was not changed at the end of the recirculation). For this reason, in the calculation of the lipase size at the interface, a lipase mass of 0.012 g was considered, corresponding to the total amount of protein present in the initial solution, and it was further confirmed by electrophoresis. The membrane emulsification process permitted controlling the dispersed phase volume that passed through the membrane and was dispersed as spherical droplets. The dispersed volume was determined upon the oil consumption from a graduated cylinder. A volume of 11.56 mL of oil was stably emulsified without phase separation. The parameters used to calculate lipase molecular radius are reported in Table 3.

The macromolecular radius of the lipase from *Candida rugosa* was 115.47 (±5) Å or 114.32 (±5) Å depending on the use of *D*[3,2] or *D*[4,3] as the mean emulsion diameter in Equations (1) and (2), respectively. Usually, information on molecular radius is obtained from crystallographic data [31,32] that allows analyzing proteins in the crystallized state. Depending on crystallization method, the following unit cell dimensions were reported: a = 64.9 Å, b = 97.5 Å, c = 175.6 Å [31] and a = 105.0 Å, b = 106.7 Å, c = 59.8 Å [32]. Obviously, when proteins are absorbed at the emulsion interface due to their amphiphilicity, they undergo conformational changes, on which the molecular ray depends, which could explain the larger size obtained in the present work. The size estimated in the present work also accounts for hindrance between adjacent macromolecules (i.e., interstitial space between spheres). Dynamic light scattering is a powerful technique in the protein crystallography that permits measuring macromolecule radiuses, likening them to a hypothetical hard sphere that diffuses with the same speed as the macromolecules under examination, but it does not permit evaluating the size of biomolecules when they are in solution and adsorbed at the emulsion interface. The unique properties of membrane emulsification have been used for the first time here to evaluate the hydrodynamic diameter of macromolecules adsorbed at droplets interface. In fact, uniform and microstructured biofunctionalized droplets are produced by a drop-by-drop mechanism, and the macromolecules are self-assembled at the interface during droplets generation. It is noteworthy that when the enzyme is distributed at the interface, it exhibits its “open” form, in which the catalytic site is accessible to the substrate and the enzyme is in its active form. This means that the simple method here proposed permits evaluating the macromolecular diameter of proteins with interfacial activity in its “open” selective conformation on the basis of easily measurable parameters: droplets diameter, dispersed phase volume, and mass of amphiphilic molecule at the interface. On the contrary, by using the conventional mechanical method, the interface is continuously constructed and destroyed by the stirring, the size of the droplets produced is not uniform, and the high-energy inputs required can modify the biomolecules’ structure.

## 4. Conclusions

Membrane emulsification permitted the formulation of stable emulsions with 10.36% oil droplets in water having a mean size of around 1.5 μm, containing at the interface 0.012 g of lipase enzyme. The macromolecular hydrodynamic diameter of lipase at the interface was around 230 Å. The method was highly reproducible provided that the following criteria apply: (i) the volume of the emulsified dispersed phase is known, (ii) the mass of the lipase at the interface is precisely measured, (iii) the size distribution of the emulsion droplets is uniform, and (iv) the emulsion is stable. The enzyme loaded at the stable emulsion interface showed very high enantioselectivity (100%) and unprecedented (*S*)-substrate conversion (100%). When the emulsion interface was produced by a mechanical stirring method, a high conversion yield was also obtained; however, the enantiomeric excess was reduced to 96% and 75% when the conversion was around 40% and 60%, respectively. Results suggested that the stable emulsions produced by membrane technology at low shear condition and without the addition of other surfactants increased the enantioselective stability of the enzyme. In this case, the enzyme specific activity was lower (accounting for the high enantioselectivity) compared to the one obtained with emulsion produced by mechanical stirring. The trend of specific activity as a function of time was more constant when the emulsion was produced by membrane emulsification. The catalytic performance of lipase measured in this work, along with the measurement of the hydrodynamic macromolecular diameter of the enzyme at the interface, demonstrated that the membrane emulsification method allows evaluating also the size of biocatalysts with interfacial activity in their catalytic enantioselective conformation, which is a key point in bioreactors construction. The membrane emulsification technology is a promising methodology to implement the application of phase transfer catalysis. Compared to classical methodology to separate enantiomers (i.e., diastereomeric crystallization), the kinetic resolution in multiphase reaction systems implemented by membrane emulsification proved great efficiency for optically pure enantiomer production.

## Figures and Tables

**Figure 1 membranes-11-00137-f001:**
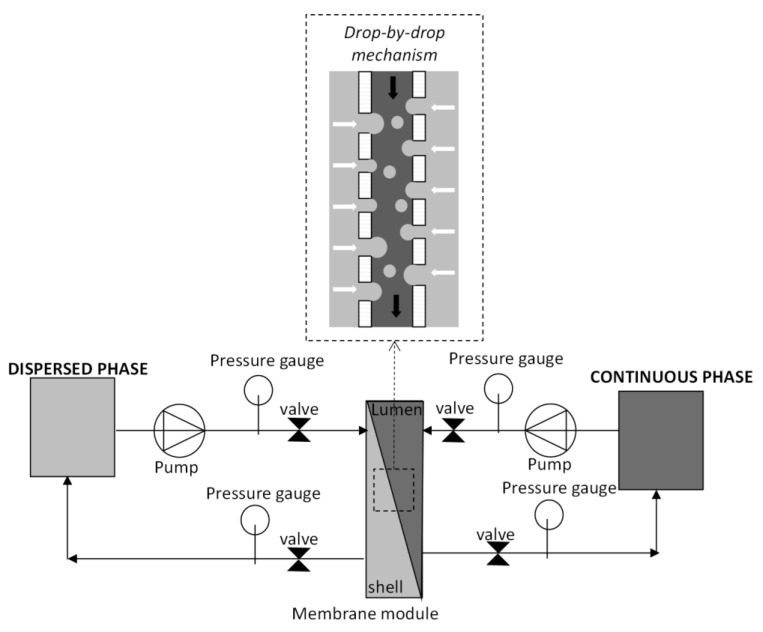
Crossflow membrane emulsification equipment and schematic representation of emulsion droplet formation at the membrane pore.

**Figure 2 membranes-11-00137-f002:**
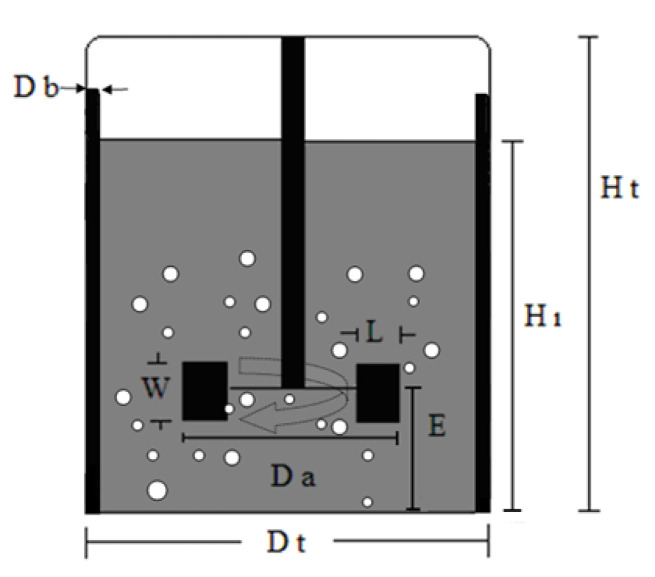
Schematic representation of stirred tank reactor geometry (impeller blade width (L), impeller blade height (W), diameter of impeller (Da), distance between middle of impeller blade and bottom of reactor (E), diameter of baffles (Db), diameter of tank (Dt), height of liquid in reactor (Hl), height of reactor (Ht)).

**Figure 3 membranes-11-00137-f003:**
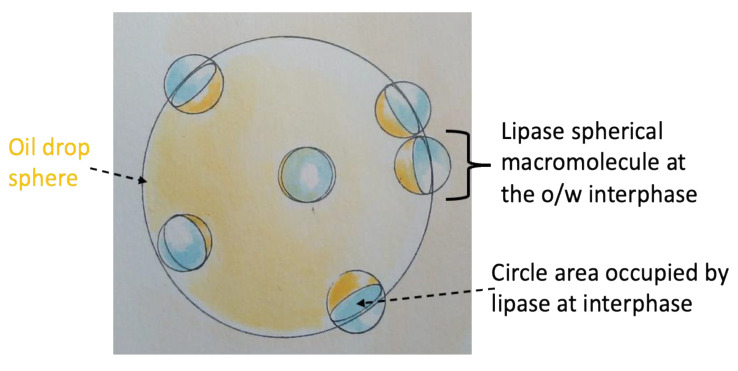
Schematic representation of lipase approximated to spherical macromolecules distributed at the oil (yellow)/water (light blue) interface of a spherical oil drop. The circle area occupied by a single lipase macromolecule is also highlighted. (To simplify the scheme, only some macromolecules from various perspectives were drawn).

**Figure 4 membranes-11-00137-f004:**
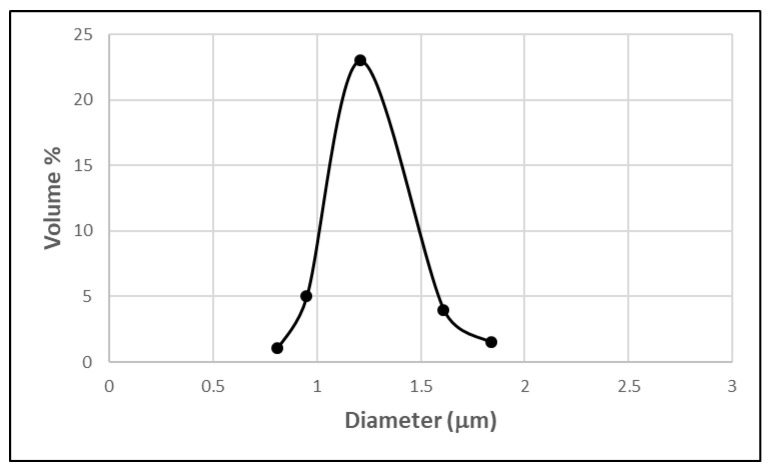
Size distribution of oil/water (o/w) emulsion produced by membrane emulsification.

**Figure 5 membranes-11-00137-f005:**
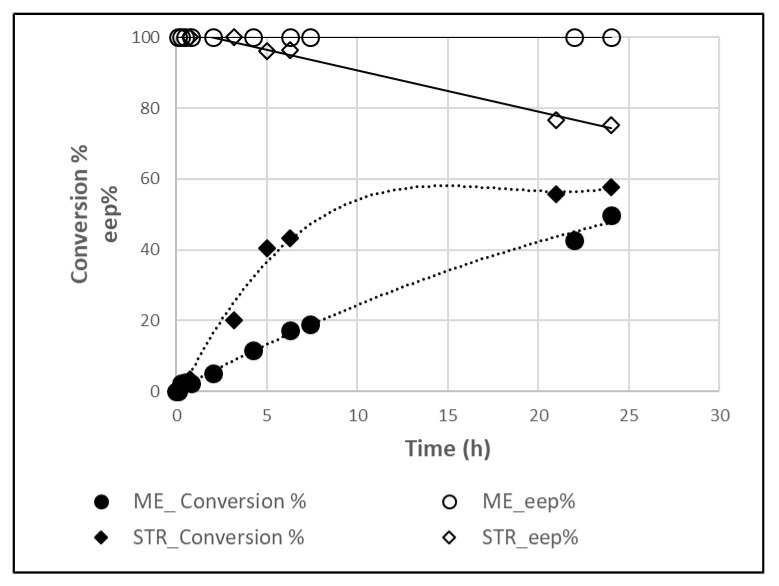
Time course of conversion and enantiomeric excess of the emulsion bioreaction carried out using the lipase distributed at the oil/water interface by membrane emulsification (ME) and mechanical stirring in a stirred tank reactor (STR).

**Figure 6 membranes-11-00137-f006:**
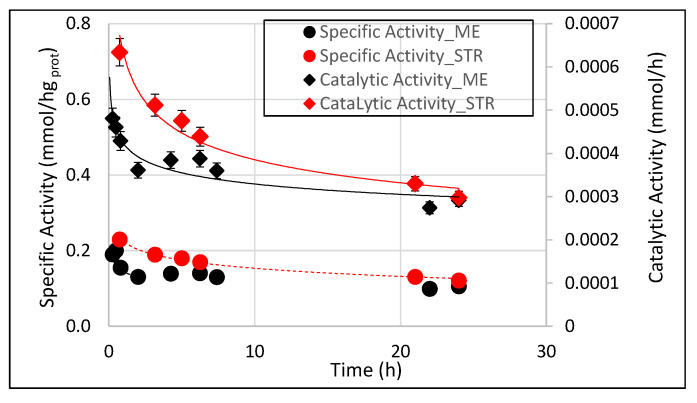
Time course of specific activity and catalytic activity of lipase distributed at the oil/water interface by membrane emulsification (ME) and stirring tank reactor (STR).

**Figure 7 membranes-11-00137-f007:**
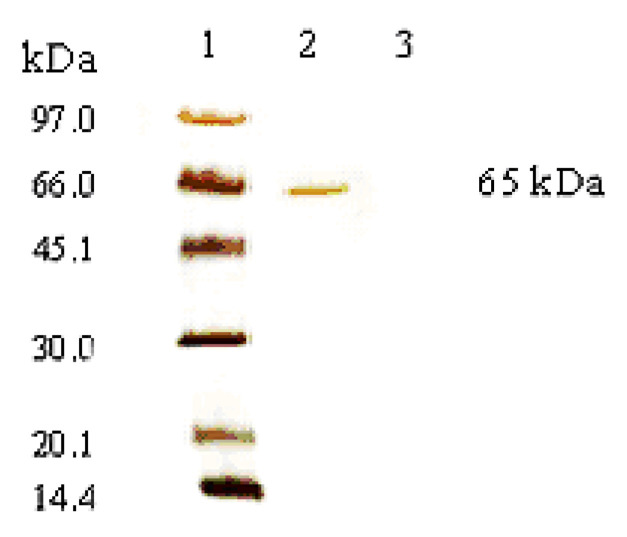
SDS-PAGE of lipase initial solution and filtered aqueous phase after emulsion preparation by membrane emulsification. Line 1: Standard solution; Line 2 represents the initial lipase solution at 2 g/L; Lines 3 represents the uncomplexed lipase present in the filtered aqueous phase of emulsion prepared using lipase solution of 2 g/L.

**Table 1 membranes-11-00137-t001:** Dimensions of stirred tank bioreactor used.

Parameter Description	Parameter Value of the Bioreactor Used	
Height of liquid in reactor (Hl) to height of reactor (Ht)	Hl = 4.8 cm	Hl/Ht = 0.8
	Ht = 6 cm	
Height of reactor (Ht) to diameter of tank (Dt)	Ht = 6 cm	Ht/Dt = 1.66
	Dt = 3.6 cm	
Diameter of impeller (Da) to diameter of tank (Dt)	Da = 1.8 cm	Da/Dt = 0.5
	Dt = 3.6 cm	
Diameter of baffles (Db) to diameter of tank (Dt)	Db = 0.3 cm	Db/Dt = 0.083
	Dt = 3.6 cm	
Impeller blade height (W) to diameter of impeller (Da)	W = 0.36 cm	W/Da = 0.2
	Da = 1.8 cm	
Impeller blade width (L) to diameter of impeller (Da)	L = 0.45 cm	L/Da = 0.25
	Da = 1.8 cm	
Distance between middle of impeller blade and bottom of reactor (E) to impeller blade height (W)	E = 0.36 cm	E/W = 1
	W = 0.36 cm	

**Table 2 membranes-11-00137-t002:** Enantiomeric excess, conversion, and specific activity measured in enantioselective hydrolysis of racemic naproxen methyl ester of lipase from *Candida rugosa* self-assembled at uniform particles interface produced by membrane emulsification (ME) and stirring method after 24 h of reaction.

Emulsification Method	Axial Velocity (m/s)	Conversion (%)	*ee_p_* (*S*) (%)	Specific Activity (mmol h^−1^ g_prot_^−1^)
ME	0.3	49.7	100	0.11
Stirring	0.9	57.8	75.2	0.12

**Table 3 membranes-11-00137-t003:** Parameters used to calculate lipase hydrodynamic diameter at the interface.

Parameter	Value
Mean particles diameter (*r_d_*) (μm)	
*D*[3,2]	1.49 (±0.3)
*D*[4,3]	1.52 (±0.3)
Total volume of the organic phase (*V_Tot_*) (mL)	11.56 (±0.5)
Volume of one drop (*V_d_*) (μL)	
from *D*[3,2]	1.73 × 10^−9^
from *D*[4,3]	1.83 × 10^−9^
Surface area of a single spherical drop (*A_d_*) (m^2^)	
from *D*[3,2]	6.97 × 10^−12^
from *D*[4,3]	7.25 × 10^−12^
Number of drops (*N_d_*)	
from *D*[3,2]	6.67 × 10^12^
from *D*[4,3]	6.29 × 10^12^
$\mathrm{Lipase}\;\mathrm{mass}\;(m_{Lipase})$ (g)	0.012
MW_lipase_ (Da)	65,000
Total number of lipase molecules at interface (*N_Tot_* _lipase_)	1.11 × 10^17^
Lipase (circle) area occupied at the interface (*A_lipase at interface_*) (m^2^)	
from *D*[3,2]	4.18 × 10^−16^
from *D*[4,3]	4.10 × 10^−16^
Lipase macromolecular radius at interface (*r_lipase at interface_*) (Å)	
from *D*[3,2]	115.47 (±5)
from *D*[4,3]	114.32 (±5)

## Data Availability

Not applicable.

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
