# Peer review of "Comparison between Lipase Performance Distributed at the O/W Interface by Membrane Emulsification and by Mechanical Stirring"

_membranes, 2021, doi:10.3390/membranes11020137_

Round 1
Reviewer 1 Report
Manuscript Number: membranes-1107924
Title: Comparison between Lipase Performance Distributed at the o/w Interface by Membrane Emulsification and by Mechanical Stirring
Article Type: Research article
Recommendation: Minor Revision
Comments to authors: The authors presented an interesting work in which membrane emulsification technology was used for the production of a microstructured emulsion bioreactor using lipase as a catalyst and as a surfactant at the same time. The work provides great insights and can be published after making the following minor changes.
- On page 9 line 335-336, author states “Data indicated that the manufacturing of emulsion interface trough the membrane technology created a favourable microenvironment”. Authors are suggested to explain those favourable microenvironment in little detail for better understanding.
- On page 9 line 344-345, author states “a constant enzyme concentration was measured (0.012 g)”. There seems to be some mistake in enzyme concentration of 0.012 g, as gram cannot be the unit of concentration.
- On page 9 line 349-350, the authors claims that “On the contrary, when the emulsion was produced by the stirring method, a continuous decrease in the specific activity was observed.”. However, on close inspection of figure 6, it does not look that there is much difference in trend of specific activity with membrane emulsification and STR
- Figure 6 shows that catalytic activity for both cases decreases with time. However, author states on page 9 that “Same trend was achieved for the catalytic activity (Figure 6) in the two systems” which indicates that catalytic activity should be constant like specific activity (when membrane emulsification was used) after ~48 mins which does not look correct.
- On page 11 line 401 author writes, “it was also verified that no lipase adsorbed to the membrane”. Authors are advised to produce more explanation on how this was verified.
Author Response
Recommendation: Minor Revision
Comments to authors: The authors presented an interesting work in which membrane emulsification technology was used for the production of a microstructured emulsion bioreactor using lipase as a catalyst and as a surfactant at the same time. The work provides great insights and can be published after making the following minor changes.
On page 9 line 335-336, author states “Data indicated that the manufacturing of emulsion interface trough the membrane technology created a favourable microenvironment”. Authors are suggested to explain those favourable microenvironment in little detail for better understanding.
Answer: The concept has been better clarified:
Pag. 9, 335-338: “Data indicated that the manufacturing of emulsion interface trough the membrane technology created a favourable microenvironment for the interaction with the hydrophobic substrate and the extraction of the hydrophilic product and a stable lipase conformation that permitted enantioselective kinetics in mild operative conditions throughout the entire reaction time.
On page 9 line 344-345, author states “a constant enzyme concentration was measured (0.012 g)”. There seems to be some mistake in enzyme concentration of 0.012 g, as gram cannot be the unit of concentration.
Answer: The mistake has been corrected and now the “enzyme amount” is used instead of “enzyme concentration.
Pag. 9, line 350-351: “After that time, a constant enzyme amount was measured (0.012 g), demonstrating that…”
On page 9 line 349-350, the authors claims that “On the contrary, when the emulsion was produced by the stirring method, a continuous decrease in the specific activity was observed.”. However, on close inspection of figure 6, it does not look that there is much difference in trend of specific activity with membrane emulsification and STR
Answer: We agree with the reviewer; a same trend is achieved by using the two methods although a more significant decrease of the specific activity was achieved during the time when the stirring method was used. This aspect has been clarified.
Pag. 9, line 355-357: “On the contrary, when the emulsion was produced by the stirring method, a continuous decrease in the specific activity decreased throughout the reaction time, even though with an initial less steep slope was observed. This is probably due to the continuous destabilization of the interface (emulsion rupture/formation) which did not permit to have a constant enzyme amount and a constant reservoir of substrate. Same trend was achieved for the catalytic activity (Figure 6) in the two systems, in fact a more severe decrease was observed in the case in which the emulsion was produced by the stirring method.”
Figure 6 shows that catalytic activity for both cases decreases with time. However, author states on page 9 that “Same trend was achieved for the catalytic activity (Figure 6) in the two systems” which indicates that catalytic activity should be constant like specific activity (when membrane emulsification was used) after ~48 mins which does not look correct.
Answer: Both the catalytic activity and the specific activity decreased with the time for both system (as already stated in the text, page 10 line 361). However, we agree with the reviewer that the sentence in which the behaviour of the specific activity achieved with membrane emulsification is potentially misleading. For that reason the text has been modified as in the following:
Page 10, line 347-349: “When the membrane emulsification method was used to produce the emulsion, first the specific activity decreased and became an almost constant specific activity (Figure 6) was observed, just after the emulsion production process was completed (about 48 min).”
On page 11 line 401 author writes, “it was also verified that no lipase adsorbed to the membrane”. Authors are advised to produce more explanation on how this was verified.
Answer: The lipase solution was recirculated in the lumen side of the membrane using a same axial velocity used for emulsion preparation for 45 min. During the time, a sample was taken from the solution vessel after 5, 15, 30, 45 min and the protein concentration was measured. Data indicated that the lipase did not adsorb on the SPG membrane surface. The procedure used has been also added in the text:
Page 11, line 409-412: “In control experiments, (i.e. by circulating a protein solution along the lumen circuit, in absence of emulsion, at the same axial velocity - 0.9 m/s - used for the continuous phase), it was also verified that no lipase adsorbed to the membrane (i.e. the lipase solution concentration was not changed at the end of the recirculation).”

Reviewer 2 Report
The authors present comparative performances of membrane emulsification technology that was used for the production of a microstructured emulsion bioreactor using lipase as a catalyst and as a surfactant at the same time, and an emulsion bioreaction system by stirring method. They critically evaluate the performance of the two methods, MS respectively STR used to distribute the enzyme at oil/water interface, considering the following parameters: enantioselectivity, conversion of the substrate, and enzyme specific activity.
The manuscript is very well structured and presented therefore I have only a few small recommendations:
- In the end of the Introduction section the authors describe the experiments that were carried out, however the purpose of the article must be clearly specified in this section.
- Small drafting mistakes; for example line 123/page 4.
- Please comment on why you choose to keep the emulsion thermostated at exactly 30 ° C in both ME and STR emulsification methods.
Author Response
Comments and Suggestions for Authors
The authors present comparative performances of membrane emulsification technology that was used for the production of a microstructured emulsion bioreactor using lipase as a catalyst and as a surfactant at the same time, and an emulsion bioreaction system by stirring method. They critically evaluate the performance of the two methods, MS respectively STR used to distribute the enzyme at oil/water interface, considering the following parameters: enantioselectivity, conversion of the substrate, and enzyme specific activity.
The manuscript is very well structured and presented therefore I have only a few small recommendations:
In the end of the Introduction section the authors describe the experiments that were carried out, however the purpose of the article must be clearly specified in this section.
Answer: The aim of the work has been further clarified in the introduction section:
Page, line : “Here we further proof the superior stability of emulsion droplets and higher enantioselectivity of lipase distributed at the interface of oil/water emulsions prepared by membrane emulsification compared to emulsions prepared by mechanical stirring. The aim of the work is to demonstrate the power of membrane emulsification technology to implement phase transfer catalysis applications.”
Small drafting mistakes; for example line 123/page 4.
Answer: We corrected the mistake
Please comment on why you choose to keep the emulsion thermostated at exactly 30 ° C in both ME and STR emulsification methods.
Answer: The temperature of 30° C has been previously demonstrated to be the optimum temperature for lipase from Candida rugosa by using different substrates. This comment has been introduced in the text:
Page 5, line 152: The emulsion was collected in a glass beaker where the reaction was continued under gentle stirrer (just to maintain a uniform dispersion of the emulsion) and thermostated at 30 °C (temperature optimum [12,13]) for 24 h.
